# Collective molecular switching in hybrid superlattices for light-modulated two-dimensional electronics

Marco Gobbi[1,6], Sara Bonacchi[1,7], Jian X. Lian[2], Alexandre Vercouter[2], Simone Bertolazzi[1], Björn Zyska[3], Melanie Timpel[4], Roberta Tatti[5], Yoann Olivier[2], Stefan Hecht [3], Marco V. Nardi[4], David Beljonne[2], Emanuele Orgiu [1,8] & Paolo Samorì [1]

Molecular switches enable the fabrication of multifunctional devices in which an electrical output can be modulated by external stimuli. The working mechanism of these devices is often hard to prove, since the molecular switching events are only indirectly confirmed through electrical characterization, without real-space visualization. Here, we show how photochromic molecules self-assembled on graphene and $MoS_2$ generate atomically precise superlattices in which a light-induced structural reorganization enables precise control over local charge carrier density in high-performance devices. By combining different experimental and theoretical approaches, we achieve exquisite control over events taking place from the molecular level to the device scale. Unique device functionalities are demonstrated, including the use of spatially confined light irradiation to define reversible lateral heterojunctions between areas possessing different doping levels. Molecular assembly and light-induced doping are analogous for graphene and $MoS_2$, demonstrating the generality of our approach to optically manipulate the electrical output of multi-responsive hybrid devices.

[1] University of Strasbourg, CNRS, ISIS UMR 7006, 8 allée Gaspard Monge, F-67000 Strasbourg, France. [2] Laboratory for Chemistry of Novel Materials, Center for Research in Molecular Electronics and Photonics, University of Mons, Place du Parc 20, 7000 Mons, Belgium. [3] Department of Chemistry and IRIS Adlershof, Humboldt-Universität zu Berlin, Brook-Taylor-Str. 2, 12489 Berlin, Germany. [4] Department of Industrial Engineering, University of Trento, Via Sommarive 9, 38123 Trento, Italy. [5] IMEM-CNR, Institute of Materials for Electronics and Magnetism, Trento unit, Via alla Cascata 56/C, Povo, 38123 Trento, Italy. [6] Present address: Centro de Fisica de Materiales (CSIC-UPV/EHU), Paseo Manuel de Lardizabal 5, E-20018 Donostia, San Sebastián, Spain. [7] Present address: Department of Chemical Sciences, University of Padua, Via Francesco Marzolo 1, Padova, 35131, Italy. [8] Present address: Institut National de la Recherche Scientifique (INRS), EMT Center, Boulevard Lionel-Boulet, Varennes, QC, J3X 1S2, 1650, Canada. Correspondence and requests for materials should be addressed to M.G. (email: marco_gobbi001@ehu.eus) or to P.S. (email: samori@unistra.fr)

One among the grand challenges of nanotechnology is the precise manipulation of an electrical output in solid-state devices through the control of molecular events occurring at the nanoscale[1]. By exploiting unique functions encoded in specific molecular groups and modulated through external stimuli, multifunctional devices can be fabricated in which the electrical conductance can be adjusted ad hoc, offering sought-after solutions for sensing and opto-electronics[2].

For instance, photochromic molecules, which are capable of switching between two (meta-) stable states when exposed to specific wavelengths[3], enable the use of a photonic input to modulate the electrical characteristics of solid-state devices[4–16]. Of particular interest is the possibility to exploit photochromic molecules to modulate the conductance of (semi)conductive materials, eventually leading to light-switchable macroscopic devices[10–16]. This approach was demonstrated for carbon nanotubes[10,11], graphene[7,12,13], and polymers[14,15].

In all these studies, the isomerization was inferred on the basis of the electrical characterization, without a direct, real-space visualization of the (supra)molecular structural changes induced by the switching events. As a consequence, the electrical effects measured at the device level could not be rationalized in various cases[12,13]. On the contrary, real-space images of supramolecular assemblies of photochromic molecules have been acquired through scanning tunneling microscopy (STM)[17–21], but either the photo-induced switching events could not be monitored[18,19] or the specific experimental conditions hampered simple translation and integration in solid-state devices[17,20,21].

Two-dimensional materials[22] (2DMs) represent an ideal platform to study the interplay between molecular assembly on surfaces and electrical transport in devices. On the exposed surface of 2DMs, well-defined molecular groups can be arranged at predetermined spatial locations with atomic precision by tailoring of supramolecular architectures[23–25]. Within these organic/inorganic superlattices, macroscopic effects taking place at the device scale can be understood on the basis of molecular functionality and nanoscale arrangement[26–30], which can be directly accessed by means of conventional surface-science techniques. Hitherto, these highly controllable superlattices have not been exploited to impart the switching properties of photochromic molecules to 2DMs.

Here we demonstrate optical control over the local charge carrier density in high-performance devices by interfacing supramolecular assemblies of photochromic molecules with 2DMs. In particular, we exploit the collective nature of self-assembly to convert single-molecule isomerization events into a spatially homogeneous switching action, which generates a macroscopic electrical response in graphene and $MoS_2$. We achieve exquisite control over such effects by combining surface-science techniques and characterization of mesoscopic devices, drawing a unified picture ranging from the scale of molecules all the way to the device. Moreover, our superlattices enable the demonstration of technologically relevant functions, such as the reversible doping in graphene and $MoS_2$, and the use of spatially confined light irradiation to pattern regions with well-defined doping levels.

## Results

**Photo-switchable hybrid superlattices**. Our approach is portrayed in Fig. 1a. The supramolecular assembly of photochromic molecules at the surface of graphene and $MoS_2$ single layers generates an atomically precise superlattice in which a major structural rearrangement is obtained by light-induced collective isomerization. As a result, the rearrangement causes a reversible shift in the 2DM work function, readable in devices as significant doping, which is also fully reversible. For this study, we designed and synthesized the spiropyran (SP) derivative bearing an 18-carbon long alkyl chain (Fig. 1a and Supplementary Note 1). SPs are photochromic molecules[31] that feature reversible photochemical isomerization between a neutral closed-ring and a zwitterionic open-ring isomer called merocyanine (MC), characterized by a larger molecular dipole. In solution, the SP→MC isomerization is triggered by irradiation with ultraviolet (UV) light, while the MC→SP back isomerization is achieved either thermally or via irradiation with visible light[31] (Supplementary Fig. 1). The long alkyl chain promotes molecular self-assembly on graphite and $MoS_2$, even at the monolayer limit[24–29,32–34]. In particular, the molecule–substrate and molecule–molecule interplay, dominated by van der Waals interactions, determines the formation of highly ordered and closely packed lamellar architectures in which alkanes adsorb flat on the surface[32,35,36] and which can be used as a template to decorate a given surface with functional groups[26,27].

In order to study the photoswitch of the SP derivative down to the monolayer limit, we performed X-ray photoelectron spectroscopy (XPS) and STM experiments at room temperature on dry films on highly oriented pyrolytic graphite (HOPG) and $MoS_2$ bulk crystals (see Methods section). Figure 1b–g shows the results obtained on HOPG, yet similar data were recorded on $MoS_2$ (Supplementary Fig. 2-3). XPS analysis of the binding energy of the N 1s core level provides unambiguous identification of the isomer on the surface, due to the very different hybridization of the indoline N in SP and MC[18]. In particular, the N 1s spectrum of spin-coated SP film (Fig. 1b) has two contributions, with a broad peak at binding energy $E_b = 406.0$ eV corresponding to $NO_2$, and a sharper feature at $E_b = 399.5$ eV to the indoline N atom. Following in situ UV irradiation, a new component appeared at $E_b = 401.0$ eV (Fig. 1c), which is a hallmark of the MC isomer[18]. Upon in situ irradiation with green light, an almost complete recovery of the initial SP spectrum was observed (Fig. 1d). These data provide evidence that photo-isomerization takes place on van der Waals substrates, in contrast with analogous experiments on metallic surfaces[18,19].

The evolution of the molecular arrangement was monitored with sub-nm resolution by STM imaging in air and at room temperature on dry films on HOPG and $MoS_2$ (see Methods section). In the SP monolayer, a lamellar structure was visualized, in which different rows of alkanes lying flat on the substrate are separated by bright fuzzy regions, generating an atomically precise superlattice (Fig. 1e). The resulting packing gives rise to a unit cell $a = 5.4 \pm 0.2$ nm, $b = 0.45 \pm 0.1$ nm, and $\alpha = 83 \pm 2°$, corresponding to an area $A = 2.5 \pm 0.2$ nm$^2$, with each unit cell containing two SP molecules. The width of the lamellae, amounting to 5.4 nm, is in good agreement with the sum of the contour length of two SP molecules, indicating that the molecules are assembling in a head-to-head fashion, as sketched in Fig. 1e, with the SP groups located in the bright regions separating adjacent lamellae. The bright contrast can be ascribed to aromatic SP head-groups, which owing to their non-planarity protrude slightly from the surface, whereas the low resolution suggests that the SP groups are not immobilized on the surface, and their dynamics occur on a time scale faster than STM imaging.

Remarkably, the UV-light-triggered SP→MC isomerization induces a profound reorganization of the assembly (Fig. 1f). In this case, the high-resolution images could be acquired throughout the whole STM field of view, indicating that fully immobilized MC groups lie close to each other, while the alkyl chains form an interdigitated lamellar structure. The packing of the MC phase exhibits a unit cell $a = 4.1 \pm 0.2$ nm, $b = 1.2 \pm 0.1$ nm, and $\alpha = 82 \pm 2°$, which still contains two MC molecules but exhibits an area $A = 4.7 \pm 0.2$ nm$^2$ that is almost twice as large as compared to the

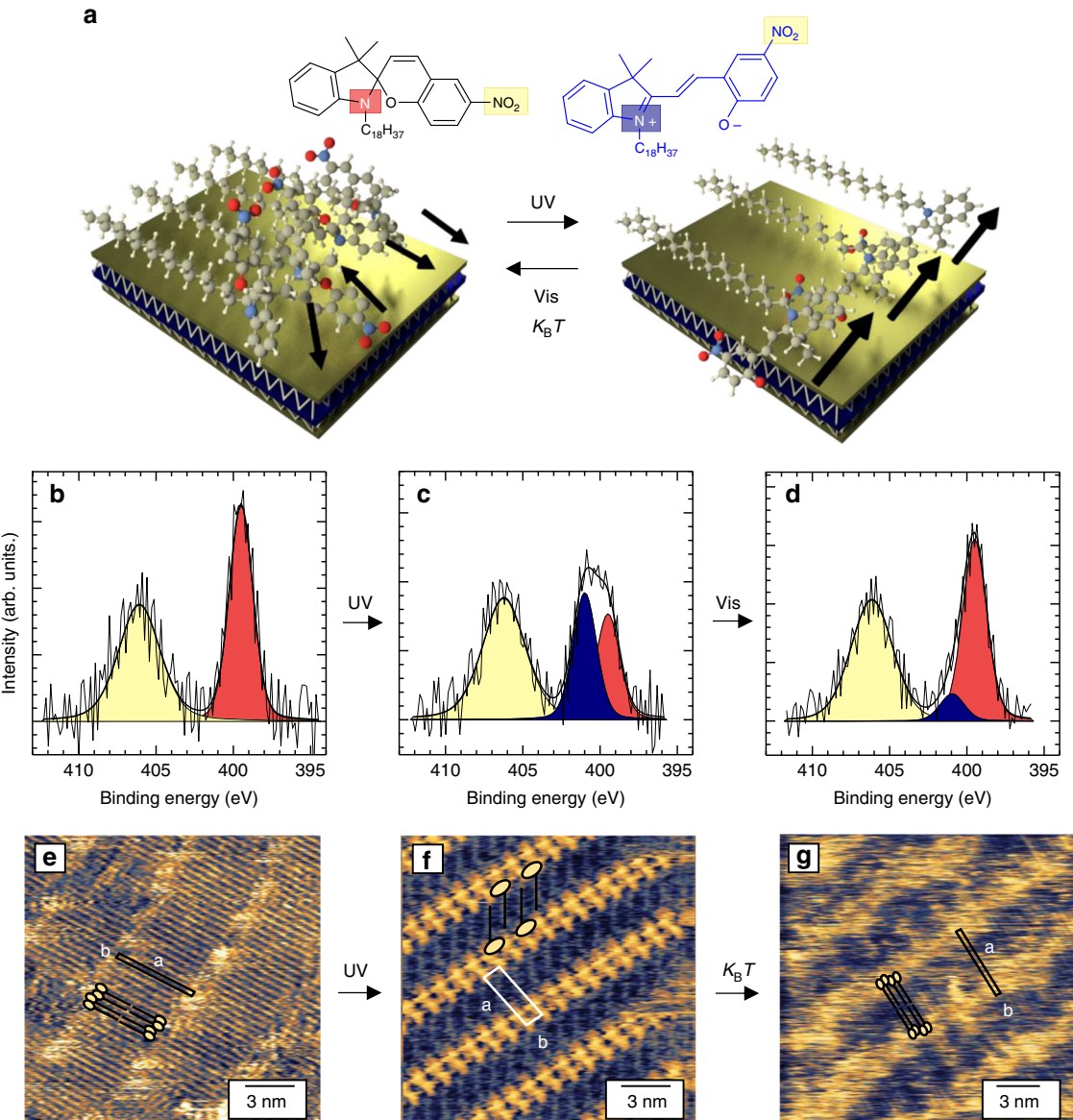

**Fig. 1** Photo-switchable molecular crystals in two dimensions. **a** Schematic representation of our approach. A spiropyran (SP) derivative forms ordered crystalline structures when deposited on different van der Waals substrates. In the cartoon, a MoS₂ single layer is depicted in which the yellow (blue) layer represents the S-(Mo-) atomic plane. Photo-induced isomerization induces a structural rearrangement. The molecular dipoles (depicted as black arrows) are randomly oriented before irradiation, yet well aligned after UV irradiation, leading to a modification in the energetics of the van der Waals substrate. The chemical structure of the spiropyran (SP)–merocyanine (MC) derivative used in the study is also shown. **b–d** N 1s core-level spectra measured on the same spin-coated ultrathin film on highly oriented pyrolytic graphite (HOPG) (**b**) kept in dark, (**c**) after in situ UV irradiation, and (**d**) after subsequent irradiation with green light. Each spectrum is characterized by multiple peaks fitted by different components, corresponding to the different N hybridization. The colors of the fitting components recall those of the N atoms in (**a**). **e–g** Scanning tunneling microscopic imaging of SP assemblies on HOPG. Height images of the molecular assemblies obtained (**e**) after spin-coating the SP solution, (**f**) immediately after UV irradiation, and (**g**) 48 h after UV irradiation. A schematic sketch of the molecule is superimposed to the images to facilitate the visualization of the molecular ordering. Tunneling parameters: $I_t = 20$ pA (**e**, **f**, **g**), $V_t = 1000$ mV (**e**, **g**), 600 mV (**f**)

SP phase (Fig. 1f). Such UV-light-induced major reorganization of the molecular assembly was also observed in survey STM images and by atomic force microscopy (AFM) at the microscopic scale (Supplementary Fig. 4). Owing to its metastable nature, at room temperature the MC isomer thermally converts back to the SP isomer in a few hours and thus gets back to its initial nanoscale arrangement. The same UV-irradiated film was imaged 48 h after irradiation, and an assembly similar to the initial one was observed (Fig. 1g), with practically identical unit cell parameters ($a = 5.4 \pm 0.2$ nm, $b = 0.4 \pm 0.1$ nm, and

$\alpha = 87 \pm 2°$). Such result is in line with spectroscopic characterization of thin films monitoring the SP→MC isomerization and subsequent thermal MC→SP recovery (Supplementary Fig. 1).

**Work function tuning through light irradiation.** Molecular mechanics/dynamics (MD) simulations combined with density functional theory (DFT) calculations made it possible to fully account for the molecular assembly as observed by STM imaging (Supplementary Note 2). In the SP case, the head-groups were found to be mobile, i.e., conformationally flexible, and hence

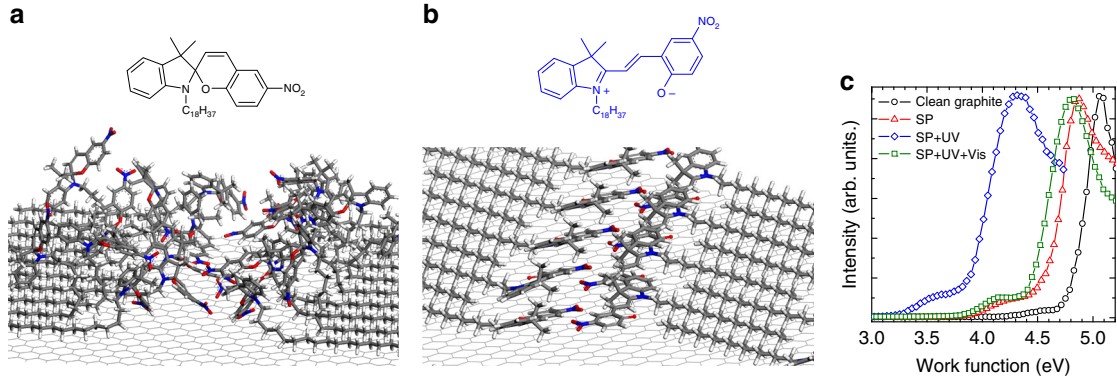

**Fig. 2** Calculated assemblies and measured evolution of the substrate work function. **a** Representative snapshot of the calculated dynamic evolution of the spiropyran (SP) assembly. **b** Self-assembly of the merocyanine (MC) isomer. The positively charged region is lifted up. **c** Evolution of the work function of highly oriented pyrolytic graphite as a consequence of the isomerization of the photochromic assembly experimentally measured by ultraviolet photoemission spectroscopy. Such evolution of the work function can be fully explained on the basis of the (average) orientation of the vertical dipoles

randomly oriented at any moment in time. A representative snapshot of such time-evolving situation is shown in Fig. 2a. On the contrary, an ordered assembly was encountered for the MC isomer, exhibiting an interdigitated structure with a unit cell in good agreement with the experimentally observed one (Fig. 2b and Supplementary Fig. 5). Importantly, the distinct assembly featured by each isomer affects the overall energetics of the underlying van der Waals layers differently. In particular, the component of molecular electric dipoles perpendicular to the surface ($\mu_z$) exerts a polarization capable of shifting the surface work function (WF) and therefore induces local doping[37]. Based on our simulations, a relatively low averaged vertical dipole was estimated for SPs ($\mu_z = 0.23$ D per molecule, Supplementary Fig. 6), since the randomly oriented dipoles of the $NO_2$ groups cancel out. In contrast, in the MC assembly the positively charged side of the molecule is lifted up for every molecule, resulting in a significant electrical dipole oriented perpendicular to the substrate plane ($\mu_z = 1.7$ D per molecule, Supplementary Fig. 5). The presence of positive vertical dipoles results in a WF reduction (or $n$-type doping), the intensity of which is proportional to the dipole magnitude.

Experimentally, the evolution of the WF of HOPG and $MoS_2$ was measured by UV photoemission spectroscopy, as a function of the switching state of the molecular layer (see Methods section). The presence of the SP film introduces a WF decrease as compared to the bare substrate, amounting to $\Delta WF = -0.2$ eV for HOPG, which is indicative of $n$-type doping (Fig. 2c). Significantly, the UV-light-triggered SP→MC isomerization causes a further, more pronounced WF decrease, $\Delta WF = -0.7$ eV for HOPG, in good agreement with the theoretical findings. Finally, the green-light-triggered MC→SP switch is accompanied by an almost complete recovery of the WF of the initial SP case (Fig. 2c). The same evolution in the WF shift was measured during the photo-isomerization of the assembly on $MoS_2$ (Supplementary Fig. 6).

**Photo-switchable electrical characteristics in hybrid devices.** The WF shift translates into a change in the electrical characteristics of devices based on single-layer 2DMs[29], offering the possibility to convert the switching of the insulating molecular crystal into a modulation in the electrical output of high-performance devices. This effect was explored in graphene and $MoS_2$ devices with pristine mobility above 5000 and 30 $cm^2 V^{-1} s^{-1}$, respectively (see Methods section). Figure 3a shows a scheme of the different steps involved in the experiment. Initially, the transfer characteristics of devices based on clean

graphene and $MoS_2$ were characterized. The measurement was then repeated after spin-coating of the SP solution and UV irradiation, which triggered SP→MC isomerization. Finally, the initial 2DM/SP superlattice conductance was recovered by irradiating the whole flake with green light. For graphene, a small shift of the charge neutrality point was observed toward negative values upon formation of the SP adlayer, corresponding to minor $n$-type doping (Fig. 3b). The SP→MC isomerization, triggered by in situ UV irradiation, introduced a significantly stronger $n$-type doping (induced electron density $n = 4.4 \times 10^{12}$ $cm^{-2}$, Fig. 3b) and the initial graphene/SP characteristics could be recovered by exposing the $MoS_2$ surface with green light (Fig. 3c). At this point, a subsequent UV light irradiation could be performed to prepare a second graphene/MC state characterized by $n$-type doping, and the whole cycle could be repeated (Supplementary Fig. 7). Similarly to the case of graphene, for $MoS_2$ the deposition of the ultrathin SP layer introduced a small shift in the threshold voltage toward negative values ($n$-type doping, Fig. 3d). The UV-light-triggered SP→MC isomerization caused a sizeable negative shift in the threshold voltage, inducing an electron density $n = 4.6 \times 10^{12}$ $cm^{-2}$ (Fig. 3d). The electrical characteristics of $MoS_2$/SP could be recovered by irradiating the entire flake with green light (Fig. 3e). Even for $MoS_2$, a second $n$-doped MC/$MoS_2$ state could be prepared by UV light irradiation, and the whole cycle could be repeated (Supplementary Fig. 7). Thanks to the non-disruptive nature of the non-covalent functionalization, the devices based on both $MoS_2$ and graphene preserved high electrical performances after the formation of the SP and MC assemblies, retaining mobilities above 25 and 4500 $cm^2 V^{-1} s^{-1}$, respectively. We also point out that the electrical characteristics of the SP/2DMs superlattices could be recovered without green light exposure by leaving the devices in dark for 24 h (Supplementary Fig. 8) in agreement with the thermal recovery observed in the optical characterization (Supplementary Fig. 1).

Numerous control experiments were performed to rule out other possible mechanisms responsible for the electrical changes induced by UV/visible light irradiation. In particular, the UV irradiation was shown to induce a shift of the electrical characteristics of 2DMs[38,39] even without photochromic molecules. In order to quantify this effect for our devices, we performed UV irradiation on the pristine devices before the formation of the SP layer. By using the same irradiation power used in the experiments with the molecular superlattices, we observed very minor effects in the case of both $MoS_2$ and graphene, as shown in Supplementary Fig. 8.

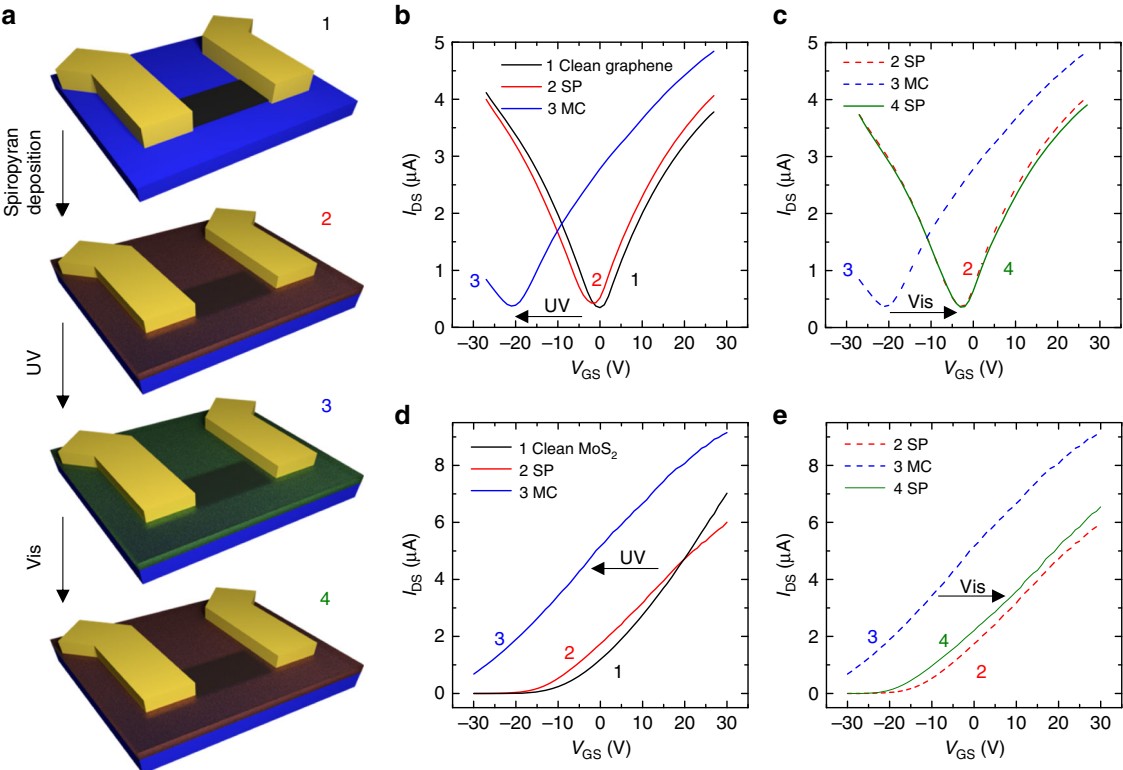

**Fig. 3** Electrical characteristics of devices based on photo-switchable superlattices. Transfer characteristics ($I_{DS}$–$V_{GS}$) of devices based on graphene and $MoS_2$ superlattices. $I_{DS}$ is the drain current, and $V_{GS}$ is the gate potential. **a** Schematics of the experiment. $I_{DS}$–$V_{GS}$ characteristics are measured on: (1) A 2DM-based clean device; (2) after spin-coating of the spiro film; (3) after irradiation with a UV light over the whole flake, which triggers the SP→MC isomerization; (4) after green light irradiation over the whole flake, which triggers the MC→SP isomerization. Gold contacts are schematically drawn in yellow, the 2DM in dark gray; the substrate in blue; the SP layer in semi-transparent red; the MC layer in semi-transparent green. **b** Trace 1 (black): clean graphene, trace 2 (red): graphene covered by the SP layer, trace 3 (blue): graphene/MC after UV irradiation over the whole flake. The arrow highlights the negative shift in the threshold voltage accompanying the SP→MC isomerization, indicative of $n$-type doping. **c** Trace 4 (green): recovered graphene/SP after green light irradiation of the whole graphene surface. Traces 2 and 3 are replotted for clarity (dashed). **d** Trace 1 (black): clean $MoS_2$, trace 2 (red): $MoS_2$ covered by the SP layer, trace 3 (blue): $MoS_2$/MC after UV irradiation over the whole flake. **e** Trace 4 (green): the pristine $MoS_2$/SP trace can be recovered upon irradiation of the whole surface with green light. Traces 2 and 3 are replotted for clarity. Details about the UV and green light irradiation are given in the Methods section; the transfer curves were measured applying a drain source voltage $V_{DS} = 10$ mV for graphene (**b**, **c**) and $V_{DS} = 100$ mV for $MoS_2$ (**d**, **e**). Channel length $L = 8.9$ μm and width $W = 1.0$ μm (graphene); $L = 1.8$ μm, $W = 3.3$ μm ($MoS_2$)

To further confirm that the strong $n$-type doping observed can be fully ascribed to the presence of the MC assembly atop, graphene and $MoS_2$ devices were covered by a layer of MC, obtained by irradiating SP in solution prior to deposition by spin-coating. In this way, the devices were not irradiated with UV light, and the effect of the MC layer could be separately addressed. Figure 4a, b show that the so-obtained MC layer introduced significant $n$-type doping in graphene and $MoS_2$, qualitatively and quantitatively similar to that observed upon direct irradiation of the SP assembly. Interestingly, in the case of graphene a dip in the electrical characteristics was observed at approximately $V_{GS} = 0$ V after the formation of the MC layer. We interpret such feature as originating from a graphene region covered by SP rather than by MC isomers. Indeed, the MC→SP back switch in solution takes place in a few seconds (Supplementary Fig. 1), and thus a significant fraction of SP molecules are spin-coated on the substrate together with the MC. In addition, we studied the nanoscale assembly obtained by directly spin-coating MC molecules on HOPG, as shown in Fig. 4c. Notably, also the directly deposited MC assembly is analogous to that obtained after UV irradiation of the dry SP film, further highlighting the direct correlation between nanoscale ordering and doping effects.

**Spatially confined modulation of charge carrier density**. Our system also allows the control over the local charge carrier density, as shown in Fig. 5 for graphene. In this case, while the UV light irradiates the whole area of the 2DM, the green light is shone only on a spatially confined region of the flake by using a focused laser (Fig. 5a and Methods section). Similarly to the case of Fig. 3, significant $n$-type doping was introduced by the SP→MC isomerization, triggered by in situ UV irradiation over the whole graphene area (Fig. 5b). Instead, the spatially confined green light irradiation resulted in a double feature in the electrical characteristics of the device (Fig. 5c), indicative of the presence of both SP and MC on the graphene surface. Indeed, the region in the graphene/MC superlattice exposed to the green laser fully recovered the almost undoped graphene/SP state, while the unexposed molecular layer maintained MC character, inducing stronger $n$-type doping. This experiment demonstrates our ability to create a dynamic heterojunction within the 2DM/photochromes superlattice. Such laser-induced modulation of charge carrier density with micrometric resolution has a high technological potential, as it might enable the realization of unconventional device architectures in which a $p$-type transistor could be reversibly converted into a rectifying $p$–$n$ junction.

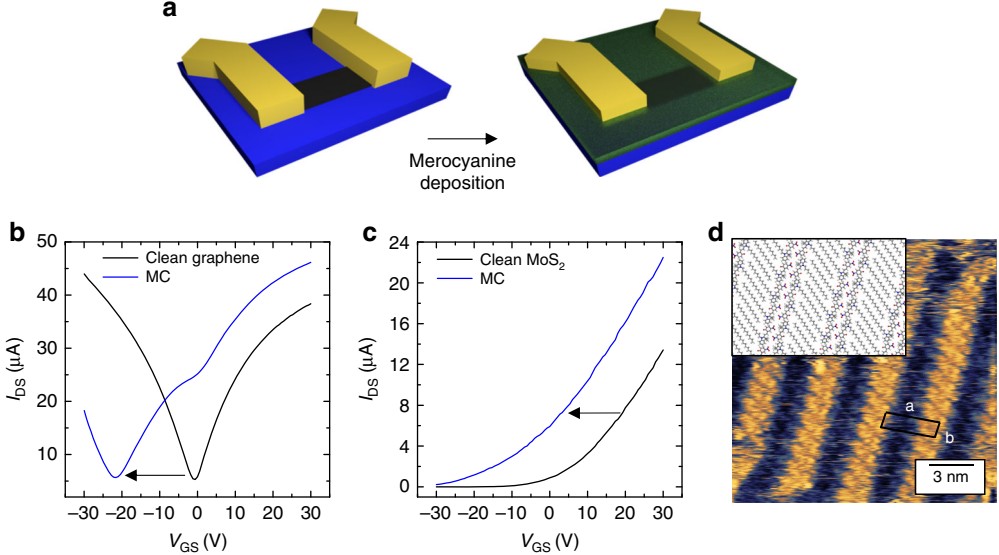

**Fig. 4** Effect of the direct deposition of the merocyanine isomer. **a** Schematics of the experiment. Transfer characteristics ($I_{DS}$–$V_{GS}$) are measured for clean graphene and MoS$_2$ devices before and after the deposition of a merocyanine film. **b** $I_{DS}$–$V_{GS}$ of a clean graphene device (in black) and of the same device covered by a directly deposited merocyanine (MC) assembly (in blue). **c** $I_{DS}$–$V_{GS}$ of a clean MoS$_2$ device (in black) and of the same device covered by a directly deposited MC assembly (in blue). The directly deposited MC layer introduced significant $n$-type doping in both graphene and MoS$_2$, qualitatively and quantitatively similar to that observed upon irradiation of the spiropyran assembly on the devices. The transfer curves were measured applying a drain source voltage $V_{DS} = 10$ mV for graphene and $V_{DS} = 1$ V for MoS$_2$. Channel length $L = 4.6$ μm and width $W = 10.6$ μm (graphene); $L = 1.4$ μm, $W = 1.0$ μm (MoS$_2$). **d** Scanning tunneling microscope height image of the assembly of a directly deposited MC film on highly oriented pyrolytic graphite. Measured unit cell parameters: $a = 3.9 ± 0.2$ nm, $b = 1.1 ± 0.1$ nm, and $\alpha = 90 ± 2°$, therefore leading to an area $A = 4.2 ± 0.2$ nm$^2$. Inset: Calculated MC assembly on the basis of molecular dynamics simulations. Tunneling parameters: average tunneling current $I_t = 20$ pA, tip bias voltage $V_t = 600$ mV. In the three cases, the SP→MC isomerization was obtained through irradiation of an SP solution with UV light immediately before spin coating

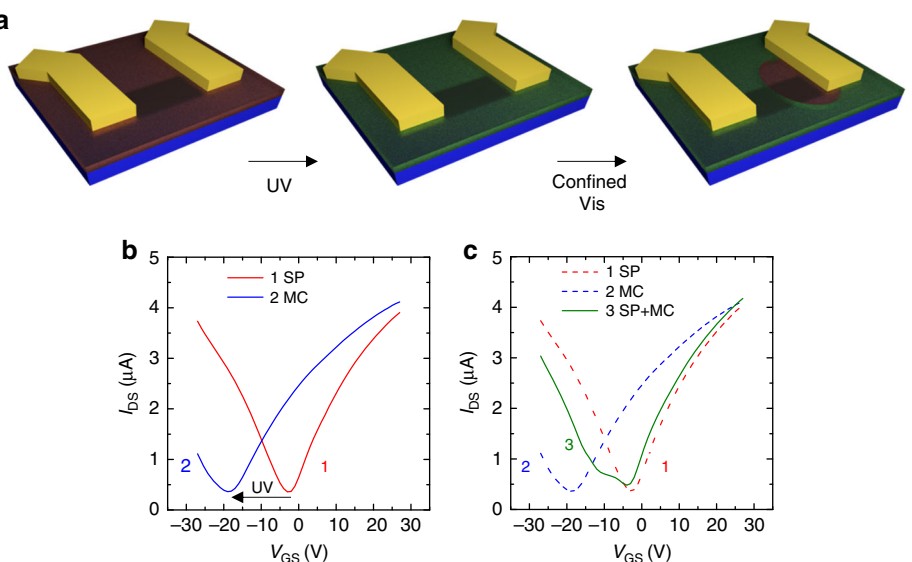

**Fig. 5** Spatially confined doping in photo-switchable superlattices. Transfer characteristics ($I_{DS}$–$V_{GS}$) of devices based on graphene and MoS$_2$ superlattices. $I_{DS}$ is the drain current, and $V_{GS}$ is the gate potential. **a** Schematics of the experiment. $I_{DS}$–$V_{GS}$ characteristics are measured for: (1) The graphene device covered by the SP film; (2) After UV light irradiation over the whole superlattice, which triggers the SP→MC isomerization; (3) After green light irradiation in a well-defined area of the active layer, which triggers the MC→SP isomerization in the irradiated area. Gold contacts are schematically drawn in yellow, graphene in dark gray; the substrate in blue; the SP layer in semi-transparent red; the MC layer in semi-transparent green. **b** Trace 1 (red): graphene covered by the SP layer, trace 2 (blue): graphene/MC after UV irradiation over the whole flake. The arrow highlights the negative shift in the threshold voltage accompanying the SP→MC isomerization, indicative of $n$-type doping. **c** Trace 3 (green solid line): graphene/MC+SP, obtained by local irradiation with green light. Traces 1 and 2 are replotted. Details about the UV and green light irradiation are given in the Methods section; the transfer curves were measured applying a drain-source voltage $V_{DS} = 10$ mV. Channel length $L = 8.9$ μm and width $W = 1.0$ μm

## Discussion

In this work, we have demonstrated multi-responsive devices relying on a collective switching action of photochromic molecules self-assembled on the surface of 2DMs. By combining different experimental and theoretical approaches, we achieve an ultra-high control over our system. An analogous control over switching events and molecular assembly was previously demonstrated in the so-called dynamic molecular crystals, i.e., self-assembled crystalline structures (typically three-dimensional) in which macroscopic structural changes arise from collective molecular events[40–42]. However, the bulk dynamic molecular crystals are typically bad electrical conductors and incompatible with macroscopic device operation, since the chemical structure of molecular switches is not designed for efficient charge transport. In this respect, our approach enables the integration of a functional supramolecular assembly, which represents the quasi-two-dimensional (2D) limit of molecular dynamic crystals, in high-performance devices to demonstrate switchable electric outputs.

The highly controllable manipulation of the charge density demonstrated in our superlattices presents several aspects of technological relevance. Precise control of the local carrier density (doping) is a key technology in semiconductor industry[43]. Most electronic devices—including diodes and metal-oxide-semiconductor field-effect transistors—are based on the possibility to generate regions within a semiconductor with spatially varying doping[43]. In our superlattices, the doping effect is tunable, reversible, and cyclable. Moreover, we demonstrated lateral heterojunctions between areas characterized by different doping levels, defined by scanning a laser light over the target areas in the superlattice. All together, these characteristics make our control over the doping in 2DMs unique. Unconventional device architectures can be envisaged, such as diodes in which the p- and n-regions can be re-defined and inverted, resulting in tunable and reversible rectification.

Our approach, relying on molecular engineering of 2DMs, is of general applicability as successfully demonstrated for both MoS2 and graphene. Therefore, the ultrathin photo-switchable molecular crystals could be integrated as an additional quasi-2D layer in vertical inorganic van der Waals heterostructures[44] with the purpose of providing a photo-responsivity with no analog in 2DMs. Our work offers a yet unexplored solution to supramolecular electronics, in which atomic precision in molecular self-assembly is tailored not to optimize charge transport but rather to control it by imparting new properties to a high-performing material, enabling the realization of multifunctional, high-performance devices.

## Methods

**Synthesis and characterization of the SP derivative**. Full details regarding the characterization and synthesis of the SP derivative are given in Supplementary Note 1.

**Optical characterization**. Absorption spectra were recorded at room temperature (25 °C) with a JASCO V-670 spectrophotometer and all solutions were examined in quartz cells with 1 cm pathlength (HELLMA) with a concentration of 0.05 mg mL$^{-1}$. The thin-film experiments were carried out in a nitrogen atmosphere.

**X-ray and UV photoelectron spectroscopy**. Surface XPS/UPS studies were performed at the BEAR endstation (BL8.1L) at the left exit of the 8.1 bending magnet of the ELETTRA synchrotron facility in Trieste (Italy). The data were collected using a hemispherical electron energy analyzer with an energy resolution of 150 meV in normal emission geometry. Photon energies of 505 and 640 eV were used for the N 1s and O 1s core levels, respectively. In this way, the kinetic energy of the emitted photoelectrons was kept at ~100 eV for each chemical species to probe similar sample depths with high surface sensitivity. For the determination of the work function, the secondary electron cutoff spectra were measured using a photon energy of 40 eV, with the sample biased at −20 V to clear the analyzer work

function. For the synchrotron measurements, the samples were irradiated in situ in ultra-high-vacuum using an optical fiber positioned close to the sample surface available at the BEAR beamline. The UV light irradiation ($\lambda = 375$ nm) was performed with power density comparable with that used for the 2D assembly isomerization described below. Instead, owing to the technical impossibility of focusing a green laser to a spot of a few micrometers at the beamline, the green light irradiation ($\lambda = 532$ nm) was carried out with significantly lower power (1 mW cm$^{-2}$) than that used for inducing the isomerization of the 2D assembly described above. Full details regarding the X-ray photoemission electron microscopy (XPEEM) images of MoS2 flakes used for device fabrication (see below) are given in Supplementary Fig. 9.

**2D assembly formation and SP-to-MC isomerization**. The 2D assemblies at the monolayer limit were obtained by spin-coating molecules from cyclohexane solutions (0.1 mg mL$^{-1}$) onto either HOPG or MoS2 bulk crystals for STM experiments and onto single-layer MoS2 and graphene devices. A mild annealing of the samples (55 °C, 30 min) after spin-coating the SP film was performed to ensure the evaporation of cyclohexane. In order to trigger the SP→MC isomerization, the so-obtained assemblies were irradiated with an UV lamp (UV-6 L/M Herolab, $\lambda = 365$ nm, power density = 1.7 mW cm$^{-2}$) for 45 min in a nitrogen-filled glovebox.

**MC-to-SP isomerization**. The MC→SP isomerization was obtained by irradiation with a green laser focused to a size of $2 \times 2\,\mu m^2$ through a 50× objective in a confocal Renishaw InVia Raman microscope ($\lambda = 532$ nm, power $P = 30\,\mu W$, exposure time $t = 0.1$ s). For flakes with size >2 μm, the laser was scanned on the flake surface irradiating a spot every 200 nm. The focused laser allowed us to perform a spatially confined irradiation of a relatively big graphene flake in a device with a channel length >8 μm by scanning the laser light only over a limited region of the flake. The samples were kept in a nitrogen-filled Linkam environmental chamber during the green irradiation.

**STM and AFM**. STM measurements were carried out by using a Veeco Scanning Tunneling microscope (multimode Nanoscope III, Veeco) operating with a piezoelectric scanner, which allowed the mapping of a maximum area of $1 \times 1\,\mu m^2$. As substrates, we used HOPG and bulk MoS2 (HQgraphene). The substrates were glued onto a magnetic disk and an electric contact was made with conductive silver paint (Aldrich Chemicals). The STM tips were mechanically cut from a Pt/Ir wire (90/10, diameter 0.25 mm). The images were obtained in air at room temperature. The raw STM data were processed through the application of background flattening, and in Fig. 1e–g in the main text the drift of the piezo was corrected using the underlying graphite lattice as a reference. The lattice of the underlying substrate was visualized by lowering the bias voltage $V_t$ to 10 mV and keeping the same average tunneling current $I_t = 60$ pA. Tip height and current were measured for all STM images.

AFM characterization was carried out in a Multimode V (Veeco) microscope equipped with a Nanoscope V controller. Commercial silicon cantilevers with a nominal spring constant of 40 N m$^{-1}$ were used for morphological characterization in tapping mode.

**Device fabrication and characterization**. Back-gated devices based on scotch-tape exfoliated graphene and MoS2 flakes were fabricated on SiO2 (90 nm)/Si substrates with a Microtech laser writer, equipped with a 405 nm laser standard photoresist (AZ1505, Microchemicals). A 35-nm-thick Au film (without adhesion layer) was thermally evaporated onto the patterned photoresist and lift-off was carried out in warm acetone (40 °C). After fabrication, the devices were immersed in warm n-methyl-2-pyrrolidone (40 °C) overnight and rinsed with acetone and isopropanol. The MoS2 devices were annealed overnight in ultra-high-vacuum at 140 °C. All devices were kept in a nitrogen-filled glove box in which they could be measured in a probe station connected to a Keithley 2636. After this procedure, the standard electrical characteristics of ideal graphene and MoS2 were measured. In particular, for the graphene device shown in Fig. 3 the charge neutrality point was found at $V_{CNP} = 0$ V, with balanced hole–electron mobility >5000 cm$^2$ V$^{-1}$ s$^{-1}$. For MoS2, the device in Fig. 3 shows the typical characteristics of MoS2 transistors, with a mobility $\mu = 32$ cm$^2$ V$^{-1}$ s$^{-1}$ and intrinsic n-type doping leading to a threshold voltage $V_T = -1$ V. The reproducibility of the results was verified by using five graphene and four MoS2 devices. In both cases, the results were in excellent qualitative and quantitative agreement with the data reported in Fig. 3.

To ensure the quality of the device fabrication protocol, a representative flake of a MoS2-based device was characterized in terms of surface chemical states and relative elemental composition via XPEEM (see Supplementary Fig. 9).

The carrier mobility was determined using the formula:

$$\mu = \left| \frac{dI_{DS}}{dV_{GS}} \right| \frac{L}{W} \frac{t}{\varepsilon_0 \varepsilon_r} \qquad (1)$$

where $L$ and $W$ are the 2DM channel length and width, $\varepsilon_0$ is the vacuum permittivity, $t$ is the thickness of gate oxide, and $\varepsilon_r$ is the relative dielectric permittivity of SiO2.

The isomerization-induced charge density was calculated as:

$$\Delta n = \varepsilon_0 \varepsilon_r \frac{\Delta V}{et} \qquad (2)$$

where $t$ is the thickness of gate oxide, $\varepsilon_0$ is the vacuum permittivity, $\varepsilon_r$ is the relative dielectric permittivity of $SiO_2$, and $e$ is the elementary charge. For graphene, $\Delta V$ was calculated as the difference in the position of the charge neutrality point before and after isomerization; for $MoS_2$, $\Delta V$ was calculated as the difference in the threshold voltage before and after isomerization.

**MM and MD simulations**. Full details regarding the MM/MD and DFT calculations calculations are given in Supplementary Note 2 and Supplementary Fig. 10.

**Data availability**. The data that support the findings of this study are available from the corresponding authors on request.

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

## Acknowledgements

We acknowledge funding from the European Commission through the Graphene Flagship (GA-785219), the Marie Sklodowska-Curie projects ITN project iSwitch (GA-642196) and IEF GALACTIC (PIEF-GA-2014-628563), and SUPER2D (GA-748971), the M-ERA.NET project MODIGLIANI, the Agence Nationale de la Recherche through the Labex projects CSC (ANR-10-LABX-0026 CSC) and NIE (ANR-11-LABX-0058 NIE) within the Investissement d'Avenir program (ANR-10-120 IDEX-0002-02), and the International Center for Frontier Research in Chemistry (icFRC) as well as the German Research Foundation (via SFB 765 and SFB 951). D.B. is a FNRS research director. M.V.N. and M.T. gratefully acknowledge the support by the CARITRO Foundation, (project MILA) Trento (Italy). Professor L. Pasquali, Dr. A. Giglia, and Dr. K. Koshmak are gratefully acknowledged for the help and stimulating discussions concerning the XPS and UPS measurements performed at the BEAR beamline at ELETTRA (proposal no. 20160339). Dr. L. Gregoratti, Dr. M. Amati, and Dr. H. Sezen are gratefully acknowledged for the help during the XPEEM sessions at ESCA microscopy beamline at ELETTRA (proposal no. 20160330).

## Author contributions

M.G., S.Bo., E.O., and P.S. conceived the experiment. B.Z. and S.H. designed and synthesized the spiropyran derivative. M.G. performed the STM study and carried out the device fabrication and characterization with the aid of S.Be. S.Bo. performed the optical characterization of the molecules. M.T., R.T., and M.V.N. performed the XPS and UPS

experiments. J.X.L., Y.O., and D.B. performed the MD/DFT calculations. M.G. and P.S. co-wrote the paper. All authors discussed the results and contributed to the interpretation of data as well as to editing the manuscript.

## Additional information

**Competing interests:** The authors declare no competing interests.

