## [Peer Review File · Nature Communications]

Reviewer #1 (Remarks to the Author):

In this manuscript, Gobbi et al. report on a monolayer device whose electrical characteristics can be reversibly switched using light. This reversible light switching is achieved by clever use of photoisomerization chemistry, which induces major structural changes in the monolayer and thereby affecting the work function of substrate. The authors provide a large amount of experimental and computational data to support their claims, and their conclusions appear sound.

While this research should certainly be published somewhere, I do not believe that Nature Communications is the appropriate venue. While this work makes elegant use of existing concepts (photoisomerization, surface dipoles, van der Waals surfaces) from the literature, I cannot see how it stimulates any new research directions for nanoscience. The authors attempt to describe the broader impact of their work in the Conclusion section (with vague references to 'demonstration of unprecedented effects' and 'molecular engineering'), however I am unconvinced that significant new research themes emerge from this work. Unless the authors can provide a very compelling rebuttal to the above, I can only recommend publication elsewhere.

I have some minor suggestions to improve the readability of the manuscript.

1. I recommend that the Introduction be expanded. The authors should write a new first paragraph, which explains the broader context of this research (i.e., address why devices with switchable electrical characteristics are desirable). This new paragraph would help to explain the broader impact of the research, and would also set the tone of the Conclusions section. The first paragraph in the current manuscript would then become the second paragraph in the new manuscript, and so on.

2. In the (current) Introduction, the important sentence at the end of the first paragraph ("...is hindered by the chemical structure of photochromic molecules, which is not designed for macroscopic device operation") is unclear to me. The authors need to give a more concrete reason as to why a 3D bulk crystal is inappropriate for macroscopic device operation.

3. Why is the long alkyl chain needed to promote self-assembly? The reasons for this are not obvious to me. While the authors make reference to a number of papers, they should describe the physical role of the alkyl chain in their own manuscript.

4. The caption of Figure 1 does not explicitly say which structure is SP, and which is MC. Moreover, the meaning of the different colors in the surface (gold, blue, white) in the substrate in Figure 1A is not explained.

5. For the sake of non-specialist readers, Figure 3 should be improved. I recommend that the authors add a diagram showing the specific configuration of their devices, and also explain the notation I_{ds} - V_{ds} . Note that the device images above the plots are not explained well in the caption at present.

6. In the MD simulation, why was force field re-parameterization necessary (i.e., what is wrong with the ordinary COMPASS for Dreiding force fields for this particular system)? How exactly was the re-parameterization carried out (i.e., was a least-squares fitting technique applied, or something else)? This re-parameterization will have major consequences on the simulation, however it is not described adequately in the Supporting Information.

7. In the evaluation of the work function shift via DFT, the authors make an important simplification by replacing the long alkyl chains with CH₃ groups. The authors need to give a rigorous, physical justification for making this approximation.

Reviewer #2 (Remarks to the Author):

In this manuscript, the authors described a way to photo-modulate electrical properties of multi-responsive hybrid devices based on 2D materials. They could achieve the same by linking inorganic surfaces with 2D supramolecular networks, in which functional groups are predictably arranged with an atomic precision. Supramolecular assembly of photochromic spiropyrans were made at the surface of graphene and MoS₂ single layers to study light-induced modulation of physico-chemical properties in a macroscopic scale. Following spiropyran (SP) → merocyanine (MC) isomerization via UV-light irradiation they could observe a pronounced decrease in work function of the underlying materials. Green-light exposure on the contrary triggered MC → SP isomerization that almost completely recover work function of the material.

Thus, they have demonstrated optical control over the local charge carrier density in high-performance transistors via a multiscale study that combines sub-nanometer resolved STM investigation and characterization of mesoscopic device. Their findings from STM imaging were in

good agreement with the outcome of molecular mechanics/dynamics simulations combined with density functional theory calculations.

A variety of analytical/spectroscopic techniques (X-ray photoelectron spectroscopy (XPS), STM absorption and emission studies, NMR etc.) have been employed to understand and characterize the molecules/materials. They could show modulation of threshold voltage in MoS₂ / graphene by irradiating UV light to the heterojunctions between inorganic surfaces and 2D supramolecular networks.

Conceptually the idea is novel and the study looks impressive. The manuscript is presented well for a wide range of scientific communities. I would recommend its publication after a minor revision.

1. The authors could show the electrical characteristics of graphene/MoS₂ could be recovered by irradiating with green light. But it is very important to study the reversibility of the system towards fatigue. Therefore, it would be of real importance if recyclability (SP-MC) of the spin-coated film is studied and included in the manuscript.

2. It is not very clear to this reviewer - whether the system under study does / doesn't encounter any leaching of the organic molecules from the inorganic layer upon isomerization of the photochromic spiropyrans.

Reviewer #3 (Remarks to the Author):

The manuscript describes how photochromic spiropyran crystals at the monolayer limit interfaced with 2D materials as successfully demonstrated for both MoS₂ and graphene for light-regulated high-performance devices. The authors provided multiscale evidence for the collective switching events occurring in an atomically precise superlattice and reversible modulation in the charge carrier density. Additionally, photo-modulation of the threshold voltage in MoS₂ and reversible lateral heterojunctions between areas possessing different doping levels were demonstrated as novel device functionalities. Hence, the results presented in this paper are of high interest and appeal for a broad scientific interest. The experiments are well conducted and the interpretation of the results is sound. The paper is written in clear style and appropriately illustrated and referenced. I am thus pleased to support its publication in nature communications.

Point-by-point reply

Reviewer #1

General comment_R1: *In this manuscript, Gobbi et al. report on a monolayer device whose electrical characteristics can be reversibly switched using light. This reversible light switching is achieved by clever use of photoisomerization chemistry, which induces major structural changes in the monolayer and thereby affecting the work function of substrate. The authors provide a large amount of experimental and computational data to support their claims, and their conclusions appear sound.*

Reply to the general comment_R1: We thank the Reviewer for appreciating our work and for acknowledging the quantity, quality, and consistency of the data presented in our work.

General criticism GC_R1: *While this research should certainly be published somewhere, I do not believe that Nature Communications is the appropriate venue. While this work makes elegant use of existing concepts (photoisomerization, surface dipoles, van der Waals surfaces) from the literature, I cannot see how it stimulates any new research directions for nanoscience. The authors attempt to describe the broader impact of their work in the Conclusion section (with vague references to 'demonstration of unprecedented effects' and 'molecular engineering'), however I am unconvinced that significant new research themes emerge from this work. Unless the authors can provide a very compelling rebuttal to the above, I can only recommend publication elsewhere.*

Answer GC_R1: We have seized opportunity to better focus our work and to more clearly highlight and explain the reasons why it is of great interest for a wider community.

Photochromic switches have previously been employed to modify the conductivity of devices. This was demonstrated by monitoring the tunnelling current flowing across individual photochromes in cumbersome single-molecule and tunnel-junction devices (refs 4–9 in the main text), or by measuring a photo-induced change in conductivity of a (semi)conductive material in contact with photochromic molecules (refs 10–16). The latter approach was demonstrated for carbon nanotubes (refs 10,11), graphene (refs 7,12,13) polymers (refs 14,15) and organic crystals. However, in all these studies, the isomerization was inferred on the basis of the electrical characterization, without a direct, real-space visualization of the (supra)molecular structural changes induced by the switching events.

To the best of our knowledge, our study represents the first in which *collective* molecular switching events are characterized with *ultra-high spatial and elemental resolution* and exploited to determine a controlled modulation in the electrical characteristics of high-performance macroscopic devices. This combination of experimental approaches provides us with an unprecedented, ultra-high control over the system under study.

In particular, we demonstrate several concepts which go well beyond the state-of-art in the field of molecules/2DMs heterostructures:

1. We exploit the **collective nature** of dynamic self-assembly to convert single-molecule switching events into a homogeneous modification of the 2DMs surface. As a consequence, the collective switching action generates a macroscopic electrical response in the 2DMs.
2. Our supramolecular approach to position specific functional groups on the 2DM surface *with atomic precision can be applied to different 2DMs*. The molecular assemblies on the surface

of both graphene and MoS₂ are analogous, so that the effects measured at the device level in both systems are qualitatively and quantitatively analogue.

3. Our ultra-thin switchable molecular layer represents the 2D limit of **dynamic molecular crystals**, and we demonstrate a way to exploit it in high-performance devices.
4. The exquisite control which we achieve over the charge carrier density in 2DMs has **profound technological implications** and enables the demonstration of novel device architectures.

We are convinced that our work will boost a **variety of further studies**. Possible research themes which can be inspired directly by our work include:

1. The study of other 2DMs and/or photochromic molecules to generate reversible light-emitting p-n junctions (for instance using WSe₂).
2. The investigation of other systems in which other photochromic molecules interact through other mechanism of doping (such as, charge transfer instead of dipolar interaction).
3. The demonstration of reversible and invert-able semiconducting p-n- diodes, in which the doping levels are defined by spatially confined light irradiation.
4. The integration of other dynamic molecular crystal in 2DM devices for switchable, high-performance electronics. Not only the optoelectronic properties, but also the magnetic properties could be targeted – for instance, by using assemblies of spin-crossover molecules.

Moreover, we are convinced that, inspired by our studies, more and more works will exploit surface science techniques to understand the working mechanism of novel devices based on 2DMs/molecules superlattices.

Our action GC_R1: We have revised and expanded the introduction and the discussion to highlight the elements of novelty of our hybrid superlattices and provide a more extensive overview on new research items which could develop from our work. In particular, the final paragraph of the introduction summarizing the experimental results (page 4) has been revised to highlight the elements beyond the state-of-art, and is reported here:

“Here, we demonstrate optical control over the local charge carrier density in high-performance devices by interfacing supramolecular assemblies of photochromic molecules with 2DMs. In particular, we exploit the collective nature of self-assembly to convert single-molecule isomerization events into a spatially homogeneous switching action, which generates a macroscopic electrical response in graphene and MoS₂. We achieve exquisite control over such effects by combining surface science techniques and characterization of mesoscopic devices, drawing a unified picture ranging from the scale of molecules all the way to the device. Moreover, our superlattices enable the demonstration of technologically relevant functions, such as the reversible doping in graphene and MoS₂, and the use of spatially-confined light irradiation to pattern regions with well-defined doping levels.”

Moreover, the technological impact of our system is highlighted in the Discussion Section (page 16-17):

“The highly controllable manipulation of the charge density demonstrated in our superlattices presents several aspects of technological relevance. Precise control of the local carrier density (doping) is a key technology in semiconductor industry⁴¹. Most electronic devices – including diodes and metal-oxide-semiconductors field-effect transistors – are based on the possibility to generate regions within a semiconductor with spatially varying doping⁴¹. In our superlattices, the doping effect is tunable, reversible and cyclable. Moreover, we demonstrated

lateral heterojunctions between areas characterized by different doping levels, defined by scanning a laser light over the target areas in the superlattice. All together, these characteristics make our control over the doping in 2DMs unique. Novel device architectures can be envisaged, such as diodes in which the p- and n- regions can be re-defined and inverted, resulting in tunable and reversible rectification.”

Minor Suggestion R1_1: *I have some minor suggestions to improve the readability of the manuscript.*

1. I recommend that the Introduction be expanded. The authors should write a new first paragraph, which explains the broader context of this research (i.e., address why devices with switchable electrical characteristics are desirable). This new paragraph would help to explain the broader impact of the research, and would also set the tone of the Conclusions section. The first paragraph in the current manuscript would then become the second paragraph in the new manuscript, and so on.

Answer R1_1: We thank the Reviewer for this suggestion. We agree that a more detailed discussion to put our research into a broader context will certainly improve the quality of the manuscript.

Our action R1_1: The introduction and discussion sections have been expanded as discussed in our reply to the general comment above. In particular, the first paragraph (page 3) now reads:

“One among the grand challenges of nanotechnology is the precise manipulation of an electrical output in solid-state devices through the control of molecular events occurring at the nano-scale¹. By exploiting unique functions encoded in specific molecular groups and modulated through external stimuli, multifunctional devices can be fabricated in which the electrical conductance can be adjusted ad hoc, offering sought-after and truly novel solutions for sensing and opto-electronics².”

Minor Suggestion R1_2: *2. In the (current) Introduction, the important sentence at the end of the first paragraph (“...is hindered by the chemical structure of photochromic molecules, which is not designed for macroscopic device operation”) is unclear to me. The authors need to give a more concrete reason as to why a 3D bulk crystal is inappropriate for macroscopic device operation.*

Answer R1_2: This comment gives us the chance to rectify the above-mentioned sentence, which could be otherwise misinterpreted. Indeed, a 3D bulk crystal would be appropriate for macroscopic device operation, but *the chemical structure of photochromic molecules* would not be. Indeed, the operation of field-effect transistors (FETs) requires efficient charge transport. The main parameter typically used for the characterization of FETs is the materials mobility (see for instance Adv. Mater. 2014, 26, 1319–1335). The chemical structure of the molecules used as active layers in organic field-effect transistors is optimized to enable charge transport with highest mobility. Instead, the chemical structure of photochromic molecules is optimized to *switch*, not to sustain charge transport. Therefore, 3D crystals of photochromes typically possess low mobility, making it unsuitable for FETs.

Our action R1_2: The analogy between our supramolecular assemblies and dynamic molecular crystals is now discussed in the “Discussion” section (on page 16):

“However, the bulk dynamic molecular crystals are typically bad electrical conductors and incompatible with macroscopic device operation, since the chemical structure of molecular switches is not designed for efficient charge transport. In this respect, our approach enables the integration of a functional supramolecular assembly, which represents the quasi-2D limit of molecular dynamic crystals, in high performance devices to demonstrate switchable electric outputs.”

Comment R1_3: 3. *Why is the long alkyl chain needed to promote self-assembly? The reasons for this are not obvious to me. While the authors make reference to a number of papers, they should describe the physical role of the alkyl chain in their own manuscript.*

Answer R1_3: We agree with the Reviewer that a short description of the physical mechanism leading to formation of the assembly could be helpful for the readers.

Well before the invention of the Scanning Tunneling Microscopy (STM), calorimetry provided experimental evidence that n-alkane assemble on graphitic substrates into a close-packed arrangement in which substrate–adsorbate interactions are maximized by arranging the molecular long axis parallel to the substrate (Everett, D. H.; Findenegg, G. H. *Nature*, 223, 52–53 (1969)). For these reasons, the assembly of alkanes on surfaces was among the first to be observed by STM (Rabe, *Science* 253, 424-427, 1991), soon becoming a model system for sub-molecular resolution studies (De Feyter, *Chem. Soc. Rev.* 32, 139-150, (2003)).

The dynamics and the nature of the interactions between alkanes and graphitic surfaces have been investigated in a number of seminal works (*Journal of Chemical Physics* 96, 6213-6221(1992); *Journal of Chemical Physics* 97, 6901-6909 (1992); *Surf. Interface Anal.* 32, 248 (2001)). It was found that: (i) the interactions between adsorbed molecules and substrate and are dominated by van der Waals forces, whereas the electrostatic interactions are very small. (ii) The adsorption stability and immobility of the alkanes increases proportionally with an increase in the chain length. In particular, adsorption energy per CH₂ group was calculated to be roughly -10 kJ mol⁻¹. (iii) The lateral molecule-molecule interactions are also dominated by van der Waals forces, and lead to the formation of closely packed lamellar structures (*Surf. Interface Anal.* 32, 248 (2001)).

In view of the high affinity of alkanes to graphitic surfaces, the alkylation of functional groups has been exploited in several occasions as a strategy to “anchor” them (non-covalently) on surfaces and promote the self-assembly (see among others: *Nanoscale* 8, 20017 (2016) and our previous work *Nat. Commun.* 8, 14767 (2017)). In our work we follow this strategy – we designed the alkylated spiropyran derivative, in which the alkyl chain does not significantly affect the molecular switching, but anchors the molecules non-covalently on the 2DM surface through van der Waals forces. In this way, the long alkyl chains facilitate the surface science studies and enable the collective reorganization which is at the basis of the macroscopic electrical effects.

Our action R1_3: We have added the above information in a new sentence on page 5:

“The long alkyl chain promotes molecular self-assembly on graphite and MoS₂, even at the monolayer limit^{24-29,32-34}. In particular, the molecule–substrate and molecule–molecule interplay, dominated by van der Waals interactions, determines the formation of highly ordered and closely-packed lamellar architectures in which alkanes adsorb flat on the surface^{32,35,36}, and which can be used as a template to decorate a given surface with functional groups^{26,27}.”

Comment R1_4: 4. *The caption of Figure 1 does not explicitly say which structure is SP, and which is MC. Moreover, the meaning of the different colors in the surface (gold, blue, white) in the substrate in Figure 1A is not explained.*

Answer R1_4: We thank the Reviewer for pointing out this issue.

Our action R1_4: The chemical structure in Fig. 1 has been labelled with SP/MC, and the caption has been changed to:

“Schematic representation of our approach. A spiropyran (SP) derivative is deposited on different van der Waals substrates. In the cartoon, a MoS₂ single-layer is depicted in which the yellow (blue) layer represents the S- (Mo-) atomic plane.”

Comment R1_5: 5. *For the sake of non-specialist readers, Figure 3 should be improved. I recommend that the authors add a diagram showing the specific configuration of their devices, and also explain the notation I_{ds} - V_{ds} . Note that the device images above the plots are not explained well in the caption at present.*

Answer R1_5: We agree that Fig. 3 could be improved for non-specialist readers.

Our action R1_5: We have separated the information contained in Fig. 3 into two separate figures (Figures 3 and 4). The device images have been substituted by a cartoon of the experiment describing in a schematic way the subsequent steps in the experiment. Minor modifications of the text were necessary to describe the new cartoons.

Comment R1_6: 6. *In the MD simulation, why was force field re-parameterization necessary (i.e., what is wrong with the ordinary COMPASS for Dreiding force fields for this particular system)? How exactly was the re-parameterization carried out (i.e., was a least-squares fitting technique applied, or something else)? This re-parameterization will have major consequences on the simulation, however it is not described adequately in the Supporting Information.*

Answer R1_6: It is common practice to readjust the parameters of classical force fields, as those do not necessarily apply to systems that differ from the training sets that have been selected for the parameterization process. This is particularly important in the case of torsion potentials and partial atomic charges involving aromatic moieties, which are often poorly described by the original force fields. The procedure to adjust the force field is now described in more details in the Supplementary Information (SI).

Our action R1_6: The procedure to adjust the force field is now described in more detail in the SI.

Comment R1_7: 7. *In the evaluation of the work function shift via DFT, the authors make an important simplification by replacing the long alkyl chains with CH_3 groups. The authors need to give a rigorous, physical justification for making this approximation.*

Answer R1_7: The approximation of replacing the long alkyl chains with CH_3 groups is motivated by two physical justifications: (i) the alkyl chains have negligible contributions to the electronic properties of the molecules, and (ii) they contribute with vanishingly small contributions to molecular dipoles compared to the polar head groups. Such a replacement is commonly performed when using DFT for complex molecules that integrate moieties that are non-functional.

Reviewer #2

General comment_R2: *In this manuscript, the authors described a way to photo-modulate electrical properties of multi-responsive hybrid devices based on 2D materials. They could achieve the same by linking inorganic surfaces with 2D supramolecular networks, in which functional groups are predictably arranged with an atomic precision. Supramolecular assembly of photochromic spiropyrans were made at the surface of graphene and MoS₂ single layers to study light-induced modulation of physico-chemical properties in a macroscopic scale. Following spiropyran (SP)→merocyanine (MC) isomerization via UV-light irradiation they could observe a pronounced decrease in work function of the underlying materials. Green-light exposure on the contrary triggered MC→SP isomerization that almost completely recover work function of the material.*

Thus, they have demonstrated optical control over the local charge carrier density in high-performance transistors via a multiscale study that combines sub-nanometer resolved STM investigation and characterization of mesoscopic device. Their findings from STM imaging were in good agreement with the outcome of molecular mechanics/dynamics simulations combined with density functional theory calculations.

A variety of analytical/spectroscopic techniques (X-ray photoelectron spectroscopy (XPS), STM absorption and emission studies, NMR etc.) have been employed to understand and characterize the molecules/materials. They could show modulation of threshold voltage in MoS₂ / graphene by irradiating UV light to the heterojunctions between inorganic surfaces and 2D supramolecular networks.

Conceptually the idea is novel and the study looks impressive. The manuscript is presented well for a wide range of scientific communities. I would recommend its publication after a minor revision.

Reply to the general comment_R2: We are thankful to the reviewer for grasping very precisely the main message of our work and for the kind words of praise.

Comment R2_1: 1. *The authors could show the electrical characteristics of graphene/MoS₂ could be recovered by irradiating with green light. But it is very important to study the reversibility of the system towards fatigue. Therefore, it would be of real importance if recyclability (SP-MC) of the spin-coated film is studied and included in the manuscript.*

Answer R2_1: We agree that a study of the recyclability of the system is important in this work. The stability of the spiro-merocyanine switching against fatigue was addressed in graphene and MoS₂ devices, by repeating UV/Vis light irradiation cycles as described in the main text. For both graphene and MoS₂, four complete cycles (including both the SP→MC and the MC→SP isomerization steps) could be performed without major degradation of the switching capability. In particular, we found that the cycles performed within one day are highly reproducible, while a deterioration of the switching is observed typically one week after the initial spin-coating. We hypothesize that the deterioration measured in the devices might be due to protonation of the SP/MC isomers (*Chem. Soc. Rev.* **43**, 148–184 (2014)). This effect could be caused by the interaction of the spin-coated SP/MC films with volatile chemicals used in the nitrogen glovebox while the sample was stored.

Our action R2_1: We included the above described new data addressing the reversibility and cyclability of our systems. In particular, the recyclability is now mentioned in the main text on page 13 while complete cycles (SP→MC and MC→SP) are shown in the new Supplementary Figure 7 on page S22 of the SI.

Comment R2_2: *It is not very clear to this reviewer - whether the system under study does / doesn't encounter any leaching of the organic molecules from the inorganic layer upon isomerization of the photochromic spiropyrans.*

Answer R2_2: The experiments described in this work are performed in the solid-state at room temperature. The solvent used for the spin-coating of the SP/MC is completely evaporated when the experiments are carried out. Indeed, we typically performed a mild annealing of the samples (55° C, 30 min) after spin-coating the SP film to ensure the evaporation of cyclohexane. We do not perform the same annealing after spin-coating the MC isomer, since otherwise it would switch back to the SP. Even in the absence of the solvent, the rearrangement in the molecular assembly accompanying the photo-isomerization is remarkable. This can be appreciated in the AFM images shown in Supplementary Figure 4, and in the STM images shown in the main text. As a result of the rearrangement, the unit cell is very much different, and the density of MC molecules on the surface as observed by STM is significantly lower than that of SP molecules. However, the molecules remain on the surface, as demonstrated by two experimental findings. First, the SP unit cell for the recovered assembly is the same (within the experimental error) as that measured for the initial SP film. Second, in the XPS measurement we did not find any difference in the ratio of the intensity of the Mo and N peak. Therefore, rather than leaching, we prefer to speak of reversible nanoscale rearrangement.

Reviewer #3

General comment_R3: *The manuscript describes how photochromic spiropyran crystals at the monolayer limit interfaced with 2D materials as successfully demonstrated for both MoS2 and graphene for light-regulated high-performance devices. The authors provided multiscale evidence for the collective switching events occurring in an atomically precise superlattice and reversible modulation in the charge carrier density. Additionally, photo-modulation of the threshold voltage in MoS2 and reversible lateral heterojunctions between areas possessing different doping levels were demonstrated as novel device functionalities. Hence, the results presented in this paper are of high interest and appeal for a broad scientific interest. The experiments are well conducted and the interpretation of the results is sound. The paper is written in clear style and appropriately illustrated and referenced. I am thus pleased to support its publication in nature communications.*

Reply to the general comment_R3: We thank the reviewer for the very positive assessment of our work and for supporting publication.

Reviewer #1 (Remarks to the Author):

The authors have provided a satisfying rebuttal to my previous criticisms. As the new manuscript makes clear, the main achievement of this work is the real-space visualization of the molecular-level structural changes which accompany photoisomerization. I agree that this is a unique achievement, and that it demonstrates several new concepts as pointed out by the authors. The revised manuscript conveys the significance of this research clearly, and for this reason I now support publication in Nature Communications.

In order to help readers place this research in context, I encourage the authors to allow for this review and rebuttal to be available online.

Reviewer #2 (Remarks to the Author):

The revised manuscript describes photo-modulation of electrical properties of multi-responsive hybrid devices based on 2D materials. The authors could reversible toggle macroscopic response in a way same as it's known for single-molecule isomerization events. Towards this, self-assembly of photochromic spyropyrans were made at the surface of graphene and MoS₂ single layers to study light-induced modulation of physico-chemical properties in a macroscopic scale. The authors provided sufficient analytical/spectroscopic evidence towards their claims.

The revised manuscript is now in a better shape. They have answered all major and minor queries in their point by point response letter as raised by the reviewers. Accordingly the manuscript is been revised and expanded.

As the outcomes are novel and the manuscript have been revised up to the satisfaction of this reviewer, I would recommend its publication.

Point-by-point reply

Reviewer #1

General comment_R1: *The authors have provided a satisfying rebuttal to my previous criticisms. As the new manuscript makes clear, the main achievement of this work is the real-space visualization of the molecular-level structural changes which accompany photoisomerization. I agree that this is a unique achievement, and that it demonstrates several new concepts as pointed out by the authors. The revised manuscript conveys the significance of this research clearly, and for this reason I now support publication in Nature Communications.*

In order to help readers place this research in context, I encourage the authors to allow for this review and rebuttal to be available online.

Reply to the general comment_R1: We thank the reviewer for the positive assessment of our work and for supporting publication. We agree that the review process has been fair and constructive, and we shall opt for the open access to the review and rebuttal letter.

Reviewer #2

General comment_R2: *The revised manuscript describes photo-modulation of electrical properties of multi-responsive hybrid devices based on 2D materials. The authors could reversible toggle macroscopic response in a way same as it's known for single-molecule isomerization events. Towards this, self-assembly of photochromic spiroopyrans were made at the surface of graphene and MoS₂ single layers to study light-induced modulation of physico-chemical properties in a macroscopic scale. The authors provided sufficient analytical/spectroscopic evidence towards their claims.*

The revised manuscript is now in a better shape. They have answered all major and minor queries in their point by point response letter as raised by the reviewers. Accordingly the manuscript is been revised and expanded.

As the outcomes are novel and the manuscript have been revised up to the satisfaction of this reviewer, I would recommend its publication.

Reply to the general comment_R2: We thank the reviewer for the positive assessment of our work and for supporting its publication.